# The Response of the Honey Bee Gut Microbiota to *Nosema ceranae* Is Modulated by the Probiotic *Pediococcus acidilactici* and the Neonicotinoid Thiamethoxam

**DOI:** 10.3390/microorganisms12010192

**Published:** 2024-01-18

**Authors:** Thania Sbaghdi, Julian R. Garneau, Simon Yersin, Frédérique Chaucheyras-Durand, Michel Bocquet, Anne Moné, Hicham El Alaoui, Philippe Bulet, Nicolas Blot, Frédéric Delbac

**Affiliations:** 1Laboratoire “Microorganismes: Génome et Environnement”, CNRS, Université Clermont Auvergne, F-63000 Clermont-Ferrand, France; thania.sbaghdi@uca.fr (T.S.); anne.mone@uca.fr (A.M.); hicham.el_alaoui@uca.fr (H.E.A.); 2Department of Fundamental Microbiology, University of Lausanne, Campus UNIL-Sorge, 1015 Lausanne, Switzerland; julian.garneau@unil.ch (J.R.G.); simon.yersin@unil.ch (S.Y.); 3Lallemand SAS, 19 Rue des Briquetiers, BP 59, CEDEX, F-31702 Blagnac, France; fchaucheyrasdurand@lallemand.com; 4Microbiologie Environnement Digestif et Santé, INRAE, Université Clermont Auvergne, F-63122 Saint-Genès Champanelle, France; 5Apimedia, BP22-Pringy, F-74371 Annecy, France; bocquetmichel@hotmail.com; 6Institute for Advanced Biosciences, CR Université Grenoble Alpes, Inserm U1209, CNRS UMR 5309, F-38000 Grenoble, France; philippe.bulet@biopark-archamps.org; 7Platform BioPark Archamps, ArchParc, F-74160 Archamps, France

**Keywords:** *Apis mellifera*, pollinators, microsporidia, gut microbiota, probiotic, neonicotinoid

## Abstract

The honey bee *Apis mellifera* is exposed to a variety of biotic and abiotic stressors, such as the highly prevalent microsporidian parasite *Nosema* (*Vairimorpha) ceranae* and neonicotinoid insecticides. Both can affect honey bee physiology and microbial gut communities, eventually reducing its lifespan. They can also have a combined effect on the insect’s survival. The use of bacterial probiotics has been proposed to improve honey bee health, but their beneficial effect remains an open question. In the present study, western honey bees were experimentally infected with *N. ceranae* spores, chronically exposed to the neonicotinoid thiamethoxam, and/or supplied daily with the homofermentative bacterium *Pediococcus acidilactici* MA18/5M thought to improve the honey bees’ tolerance to the parasite. Deep shotgun metagenomic sequencing allowed the response of the gut microbiota to be investigated with a taxonomic resolution at the species level. All treatments induced significant changes in honey bee gut bacterial communities. *Nosema ceranae* infection increased the abundance of *Proteus mirabilis*, *Frischella perrara*, and *Gilliamella apicola* and reduced the abundance of *Bifidobacterium asteroides*, *Fructobacillus fructosus*, and *Lactobacillus* spp. Supplementation with *P. acidilactici* overturned some of these alterations, bringing back the abundance of some altered species close to the relative abundance found in the controls. Surprisingly, the exposure to thiamethoxam also restored the relative abundance of some species modulated by *N. ceranae*. This study shows that stressors and probiotics may have an antagonistic impact on honey bee gut bacterial communities and that *P. acidilactici* may have a protective effect against the dysbiosis induced by an infection with *N. ceranae.*

## 1. Introduction

The holobiont is defined as the functional association of a pluricellular host and its associated microbial communities [1,2]. It is becoming increasingly clear that the host has a significant impact on the microbiota composition and vice versa. For instance, the gut microbiota can affect animal neurodevelopment and behavior and may even have effects on the collective behaviors of social organisms, as is suggested for honey bees [3,4]. Honey bee microbiota assists its host in essential functions, including the digestion of nutrients [5]. It supports general honey bee health and is involved in bee immunity, such as resistance to pathogens [6,7,8,9,10,11].

The composition of the honey bee gut microbiota is considered to be relatively simple, as it is represented by only five to nine bacterial phylotypes [8,12]. The majority of the core microbiota includes two proteobacteria: *Snodgrassella alvi* and *Gilliamella apicola*, with the latter separated into two species: *Gilliamella apicola* and *Gilliamella apis* [13]. It also includes the actinobacterium *Bifidobacterium* and two Lactobacillus-related clusters: *Lactobacillus* Firm-4, now referred to as *Bombilactobacillus* spp. (with *B. mellis* and *B. mellifer)*, and *Lactobacillus* Firm-5, corresponding to the amended *Lactobacillus* genus (with *L. apis*, *L. helsingborgensis*, *L. kimbladii*, *L. kullabergensis*, and *L. melliventris*) [14,15]. The proteobacteria *Frischella perrara*, *Bartonella apis*, and *Parasaccharibacter apium*, as well as a *Gluconobacter*-related species group known as Alpha2.1, are also members of the gut microbiota but are less abundant in proportion and usually have a lower prevalence [8]. They are considered rare members but participate in the composition of the dominant set of the honey bee gut microbiota. Although other non-core species are able to colonize the honey bee, these less frequently observed taxa are thought to appear in bees that have been stressed or highly exposed to invasive colonizers in the environment [16,17].

As a holobiont, the honey bee is exposed to a variety of biotic and abiotic stressors that affect both the insect and its gut microbiota [10,18,19]. Among the biotic threats, the microsporidia *Nosema ceranae* is a highly prevalent intestinal parasite known to affect the gut microbiota [12,20,21,22]. This ubiquitous parasite is transmitted by its spores via the feco-oral route between bees, particularly during cleaning activities [23]. *Nosema ceranae* infection is known to significantly increase the relative abundance of *G. apicola* and *S. alvi* and to decrease that of *Alphaproteobacteria*, *Bifidobacterium* spp., and *Lactobacillus* spp. Such dysbiosis could sensitize the honey bee to opportunistic pathogens [20,21].

The gut microbiota may also be impacted by abiotic threats like pesticides and especially neonicotinoid insecticides. Neonicotinoids are widely used for pest management but unfortunately also affect non-target insects like honey bees [24,25]. A chronic exposure to thiacloprid transiently reduced the absolute abundance of *Bombella apis* and *Lactobacillus* Firm-5 [26]. Both imidacloprid and thiamethoxam reduced the relative abundance of *Lactobacillus* spp. regardless of the season [22]. The thiamethoxam metabolite clothianidin was found to cause dysbiosis in the gut microbiota [27].

The use of endogenous and exogenous bacterial strains as potential probiotics has been suggested as a strategy to enhance honey bee health by mitigating the detrimental effects on the gut microbiota of various pathogens, including *N. ceranae* [28,29], although their mode of action remains to be determined. A higher survival, a decrease in *N. ceranae* proliferation, and an increase in honey bee population in colonies were reported when supplementing with Protexin^®^ (ADM Protexin Ltd., Somerset, UK), a formulation based on *Enterococcus faecium* [30,31]. The lactic acid bacterium *Pediococcus acidilactici* appeared to improve the survival of honey bees that were infected by *N. ceranae* and/or exposed to other compounds [32,33], suggesting that it can be used as a potential probiotic.

This study aimed to decipher the impact of *N. ceranae* and thiamethoxam, alone or in combination, on the honey bee gut microbiota and to test whether *P. acidilactici* was able to counteract some alterations. Honey bee workers were infected with *N. ceranae* spores, orally exposed to a low dose of thiamethoxam, and treated with *P. acidilactici* alone or in combination. The modulations of the gut microbiota were studied at the species level using whole-genome shotgun metagenomics.

## 2. Materials and Methods

### 2.1. Honey Bee Rearing and Experimental Procedures

*Apis mellifera* (genotype Buckfast) workers were sampled from brood frames from three different colonies (Col1, Col2, and Col3) of the same apiary (UMR 6023, Clermont Auvergne University, Clermont-Ferrand, France) in September 2021. They were anesthetized with CO_2_ and cohorts of 80 worker bees were distributed in Pain-type cages containing a 5 mm piece of TempQueen to mimic five queens’ mandibular pheromones (Intko Supply Ltd., Chiliwack, BC, Canada) and maintained at 35 °C. Honey bees were fed ad libitum with 50% sucrose syrup (*w*/*v*) complemented with 1% (*w*/*v*) protein supplement (Provita’Bee, ATZ Dietetic, Mas-Cabardès, France). Seven experimental conditions were tested for each colony replicate: (1) untreated control (C), (2) treatment with *P. acidilactici* (P), (3) chronic exposure to 1.5 ng/mL thiamethoxam (T), (4) infection by *N. ceranae* (N), (5) infection by *N. ceranae* then treatment with *P. acidilactici* (NP), (6) infection by *N. ceranae* then chronic exposure to 1.5 ng/mL thiamethoxam (NT), and (7) chronic exposure to 1.5 ng/mL thiamethoxam and treatment with *P. acidilactici* (TP). *Nosema ceranae* spores were freshly purified from experimentally infected honey bees using a Percoll gradient to obtain 99.9% purity [34]. *Nosema ceranae*-infected groups were collectively infected by adding ca. 10^5^ spores per bee in sugar syrup [21]. When the feeders containing *N. ceranae* spores were empty (within 12 h), they were removed and replaced with feeders containing only sucrose syrup. The homofermentative *Pediococcus acidilactici* MA18/5M (Bactocell^®^, Lallemand SAS, Blagnac, France) was grown overnight in MRS medium at 37 °C. The cell concentration was determined by measuring absorbance at 600 nm (absorbance value of 1 at 600 nm ≈ 1.6 × 10^8^ CFU/mL) and the sucrose syrup was supplemented daily with bacteria for a final concentration of 10^4^ CFU/mL. Honey bees were fed ad libitum with sucrose syrup supplemented or not with thiamethoxam and/or *P. acidilactici* based on their experimental condition. For the TP condition, thiamethoxam and *P. acidilactici* were administered in different feeders at 1.5 ng/mL and 10^4^ CFU/mL, respectively. The feeders were changed and sucrose consumption and mortality were monitored daily. All treatments were repeated for the three colonies (i.e., replicates).

### 2.2. Sampling, DNA Purification, and Shotgun Metagenomic Sequencing

Sixteen days after treatment initiation, six whole guts, excluding crops, were dissected and pooled in 1 mL PBS containing ca. 0.3 g of 1 mm glass beads. The absence or presence of *N. ceranae* spores in the guts was confirmed by bright field microscopy. To reduce the quantity of eukaryotic DNA from both the host and the parasite, bacterial enrichment was performed as previously described [35]. DNA was extracted using a phenol-chloroform method. DNA was quantified using Qubit^TM^ 3.0 Fluorometer (dsDNA HS assay kit, Thermo Fisher Scientific Inc, Waltham, MA, USA) and 200 ng total DNA were randomly fragmented by sonication using a Covaris E220 (Covaris^®^, Woburn, MA, USA) with a target fragment length of 500 bp. Paired-end libraries were prepared using the Illumina TruSeq^®^ DNA Nano library kit. Libraries were sequenced to obtain 2 × 150 bp paired-end reads using a NovaSeq 6000 apparatus (Illumina^®^, San Diego, CA, USA).

### 2.3. Data Processing and Taxonomical Analysis

Sequencing read quality control was performed using *fastp* with default parameters [36]. Reads from *A. mellifera* and *N. ceranae* were removed from the dataset using bowtie2 [37]. We also verified the absence of *N. apis* by mapping all the reads on a 382 bp fragment of the *N. apis* gene encoding the small subunit ribosomal RNA. No read mapped this region. All the samples were randomly subsampled to 3.88 million reads based on the sample with the fewest reads. The relative abundance of microbial communities was assessed using MetaPhlAn4 (Version Jan 21) with default parameters [38]. Data were filtered by keeping species that met at least one of the following criteria: (i) The species was present in at least one sample with a relative abundance higher than 0.1%, (ii) the species had a relative abundance lower than 0.1% but was present in at least half of the samples, or (iii) the species had a relative abundance lower than 0.1% but was present in all the replicates of a same condition. Significant biomarker and discriminant feature discovery was performed using LEfSe with default parameters using a linear discriminant analysis (LDA) score ≥ 2 between conditions [39]. Principal component analyses (PCA), biplots, and other graphs were generated using R packages *ggplot2* (Version 3.3.6), *corrplot* (V. 0.92), *FactoMineR* (V. 2.7), *factoextra* (V. 1.0.7), and *rgl* (V. 0.110.2). The raw data were deposited in NCBI Sequence Read Archive with BioProject accession number PRJNA977416.

### 2.4. Statistical Analysis

Statistical analysis was performed using Prism (Version 10.0.3) statistical software. Honey bee survival was analyzed using the Kaplan–Meier method followed by the log-rank Mantel–Cox test. The pairwise Mann–Whitney U test was used to compare the infection levels between experimentally infected and not experimentally infected groups.

## 3. Results

### 3.1. Honey Bee Survival in Response to Treatments

Honey bee workers were infected with *N. ceranae*, chronically exposed to a solution of 1.5 ng/mL of thiamethoxam, and/or fed with the lactic acid bacterium *P. acidilactici* (10^4^ CFU/mL). The previous treatments were applied alone or in combination. None of the treatments led to significant differences in syrup consumption. Honey bees consumed, on average, 368 CFU/bee/day of *P. acidilactici* and 0.1 ng/bee/day of thiamethoxam.

Honey bees that were not experimentally infected (considered control bees) exhibited a natural infection with *N. ceranae*, with a prevalence of infected bees of 17.6 ± 10.1%, 25.6 ± 15.4%, and 35.7 ± 23.4% for colonies 1, 2, and 3, respectively. Experimentally infected honey bees showed a significant increase in parasite prevalence (*p*-value > 0.0001, reaching almost 100% of infected honey bees (Appendix A).

Infection with *N. ceranae* significantly reduced survival of the honey bees, whether they were exposed to thiamethoxam or not, compared to uninfected control worker bees (Figure 1). Thiamethoxam and *P. acidilactici* administered alone did not significantly affect honey bee survival. Thiamethoxam associated with *P. acidilactici* led to a significant decrease in survival (*p*-value = 0.0021). In contrast, introducing *P. acidilactici* to infected honey bees appeared to boost overall survival (82.2% of survival in NP *versus* 78.2% in N at day 16), although the difference was not statistically significant (Appendix A).

### 3.2. More Than Half of the Gut Microbiota Was Represented by Bartonella apis

The mean proportion of total reads assigned to bacteria in all samples was 76.5 ± 3%. Fifty species were eventually identified, belonging to 31 genera, 20 families, 12 orders, 7 classes, and 3 phyla. The gut microbiota composition from untreated bees was assessed using the mean values of the three replicates from the control group (C) (Figure 2A). The core microbiota, described in former studies [40,41], represented 14.1 ± 2.5% of the total relative abundance. It included *Snodgrassella alvi* (2.41 ± 0.85%), *Gilliamella apis* (3.39 ± 2.8%), *Gilliamella apicola* (1.38 ± 0.18%), *Lactobacillus apis* (1.99 ± 0.55%), *Lactobacillus melliventris* (1.60 ± 0.96%), *Lactobacillus helsingborgensis* (0.97 ± 0.90%), *Lactobacillus kullabergensis* (0.20 ± 0.08%), *Lactobacillus kimbladii* (0.13 ± 0.06%), *Bombilactobacillus mellis* (0.22 ± 0.09%), *Bombilactobacillus mellifer* (0.02 ± 0.01%), and *Bifidobacterium asteroides* (1.84 ± 0.16%). Non-core bacteria represented a high proportion of the microbiota (64.0 ± 4.8%). In this study, the Alphaproteobacteria *Bartonella apis*, recently described as a gut symbiont [42], was the most abundant species (56.48 ± 6.1%). Other species were less abundant on average, such as *Frischella perrara* (1.42 ± 0.9%) and *Apilactobacillus kunkeei* (1.46 ± 0.6%).

Considering all experimental conditions, *Bartonellaceae* and *Lactobacillaceae* were the most abundant families (56.6 ± 0.8% and 4.1 ± 1.5%, respectively, Figure 2B). *Bartonella* and *Gilliamella* were the most abundant genera (56.6 ± 0.8% and 5.2 ± 2.4%, respectively). The most abundant species identified in all samples, regardless of the condition, was *Bartonella apis* (53.9 ± 6.5%) followed by *S. alvi* (3.1 ± 1.5%). The comparison of species abundance showed a high increase in *Proteus mirabilis* in infected honey bee gut (1.95 ± 1.03%) in comparison to the control group (0.10 ± 0.03%). Some species appeared to be present only in some samples, such as *Arsenophonus* spp., which represented 7.8% in one sample (TP2, bees co-exposed to thiamethoxam and *P. acidilactici*), whereas it was undetected in all other samples (mean of 0.02 ± 0.04%).

PCA based on the relative abundance of bacterial species revealed a separation of the infected group (N) from the control group (C), *N. ceranae* and thiamethoxam (NT), and *N. ceranae* and *P. acidilactici* (NP) (Figure 3). Furthermore, the samples of group N appeared to be closer to each other compared to the other treatments, suggesting a lower within-group variability. Of note, pairwise comparisons showed no significant differences in alpha and beta diversity indices between the tested conditions.

### 3.3. N. ceranae Altered the Abundance of Some Gut Bacterial Species

PCA based on the relative abundance of species seemed to segregate control (C) and infected (N) samples, with *F. perrara*, *G. apicola*, and *P. mirabilis* strongly contributing to the clustering towards the infected group, whereas *L. kimbladii*, *B. asteroides*, and *Bombilactobacillus* spp. leaned toward the non-infected group (Figure 4A). LDA analysis highlighted the significantly enriched or depleted bacterial species in infected honey bee guts. The relative abundance of *P. mirabilis* was 20 times higher in infected honey bees (1.95 ± 1.03%) compared to uninfected ones (0.10 ± 0.03%). The abundance of *G. apicola* (2.02 ± 0.25% and 1.38 ± 0.18% in N and C, respectively) and *F. perrara* (2.75 ± 0.44% and 1.42 ± 0.89%) was also significantly increased in infected guts. In contrast, infected honey bees were significantly depleted of lactic acid bacteria (LAB), namely, *B. asteroides*, *L. kimbladii*, *L. kullabergensis*, *L. melliventris*, *B. mellis*, *B. mellifer*, and *F. fructosus*, with a decrease in relative abundance ranging from 2.3 times to 18.4 times compared to the control (Figure 4B).

### 3.4. Pediococcus acidilactici Attenuated the Modulation of Specific Lactic Acid Bacteria by Nosema ceranae

The addition of *P. acidilactici* to the syrup had no significant effect by itself on the composition of the gut microbiota in uninfected bees except for *P. mirabilis* (Appendix A). In contrast, several species contributed to the separation of the infected samples (N) and the infected *P. acidilactici*-treated (NP) groups (Figure 5A). LDA indicated that *B. asteroides*, *Klebsiella oxytoca*, *B. mellifer*, and *B. mellis* were significantly increased when infected bees were fed with *P. acidilactici* (NP vs. N, Figure 5B). Two species, *B. apis* and *Gottschalkia acidurici*, were depleted (not shown for *G. acidurici*, as it was undetected in the NP group). The comparison of infected honey bees fed with *P. acidilactici* (NP) and the control group (C) showed that three species were more abundant in the NP group: *P. mirabilis*, *G. apicola*, and *Morganella morganii*, and one species was 14-fold less abundant: *Lactobacillus kimbladii* (Appendix A).

### 3.5. Thiamethoxam Also Counteracted Specific Alterations Induced by N. ceranae

The relative abundance of only one species, *B. asteroides*, was affected by the chronic and sublethal exposure to thiamethoxam, with a significant decrease compared to the control group (Appendix A). PCA analysis showed that LAB species contributed to the separation of the N and NT groups (Figure 6A). A significant increase in the abundance of *L. kullabergensis*, *A. kunkeii*, *B. mellis*, *B. mellifer*, *B. coryneforme*, and *B. asteroides* and a significant decrease in *F. perrara* were observed in infected honey bees exposed to thiamethoxam (NT) compared to the unexposed group (N, Figure 6B). Interestingly, the abundance of these species did not significantly differ compared to the untreated control group (NT vs. C), suggesting that thiamethoxam counteracted the effect of *N. ceranae* for these species. Only two species showed different abundance between the infected and thiamethoxam-exposed group (NT) and the control group (C): *P. mirabilis* was more abundant and *L. kimbladii* was five times less abundant in the NT group (Appendix A).

## 4. Discussion

The taxonomic assignation based on numerous genomic markers allowed for unambiguous assignments at the species level and for a more accurate estimation of relative abundance in comparison to the DNA metabarcoding approach based on *16S* rRNA gene [38]. For example, it allowed the response of the different *Gilliamella* and *Lactobacillus* species to be distinguished.

The taxonomic microbial composition of the untreated bees (C group) differed from most of those previously reported, which showed a large domination by *Lactobacillus* and *Bombilactobacillus* genera followed by *Gilliamella* spp. and *Snodgrassella alvi* [8]. In this work, the bee guts were largely dominated by *Bartonella apis*. A high level of *Bartonella* had been reported in overwintering honey bees and was associated with a lower abundance of the core bacterial taxa [43,44]. It has been proposed that this shift could be due to pollen shortages since *B. apis* is able to use alternative energetic sources such as lactate and ethanol [44]. Castelli et al. have shown that nutritional stress (i.e., monofloral vs. polyfloral pollen) could also increase the relative abundance of *B. apis* [45]. Here, honey bees were collected at the transition between the summer and winter population, which can potentially explain the high abundance of *B. apis.* However, a geographic and/or ecotypic variation of the gut microbiota cannot be excluded, as suggested by [46].

Infection with *N. ceranae* significantly decreased the survival of the honey bees; however, this varied between colonies (Figure 1 and Appendix A). Previous studies have shown that *N. ceranae* can induce a significant decrease in survival [32,33,47] but not consistently [48,49]. The presence of *N. ceranae* increased the relative abundance of species belonging to the *Proteobacteria* phylum and decreased the relative abundance of those belonging to *Firmicutes* and *Actinobacteria* in comparison to the C group. This could be seen as a complete displacement of the gut microbiota [50].

*N. ceranae* might indirectly impact the microbiota by changing the morphology and the homeostasis of the bee gut. For instance, *F. perrara*, which was significantly increased in presence of *N. ceranae*, is known to trigger a morphological change in the epithelial surface in the gut by inducing scab formation that could be linked to immune responses [51]. *N. ceranae* can modulate the expression of immune-related genes, including genes coding for antimicrobial peptides (AMPs) [33,52,53,54,55], which could eventually impact the gut microbial communities. In return, the microbiota modulations following infection by *N. ceranae* could affect gut functions. The core species that increased (*G. apicola* and *S. alvi*) or decreased (*Lactobacillus* and *Bombilactobacillus* spp.) in response to infection are thought to participate in digestive functions by breaking down complex sugars and secreting fermentation products [8,56]. Such modifications could thus alter the metabolites available for the host as well as its sensitivity to other pathogens. For instance, some lactic acid bacteria strains can produce antimicrobial molecules [8]. *P. mirabilis*, whose relative abundance increased in infected honey bees, is thought to have beneficial effects against *Paenibacillus larvae*, the causative agent of American foulbrood [57].

The supplementation of infected honey bees with *P. acidilactici* seemed to partially restore their gut microbiota (Figure 5B), suggesting that it can act as a potential probiotic but with limited improvement in survival (Figure 1). Although *P. acidilactici* impacted some bacterial species in honey bees infected with *N. ceranae*, *P. acidilactici* itself was not detected. This suggests that the bacteria did not colonize or proliferate in the gut. *P. acidilactici* would restore the microbiota indirectly by producing modulating compounds rather than via direct competition. In broilers, a symbiotic supplement containing *P. acidilactici* reduces the amount of *Clostridium perfringens* in the intestine and raises *C. perfringens* antibodies in the mucosa, which lessens the severity of necrotic enteritis [58]. In addition to restoring the gut microbiota, *P. acidilactici* can also reestablish the expression of genes encoding catalase and glutathione peroxidase, which were found to be significantly reduced by *N. ceranae* [33], suggesting a potential probiotic effect on the honey bee holobiont.

Thiamethoxam also appeared to mitigate the dysbiosis caused by *N. ceranae* except for *P. mirabilis* and *L. kimbladii* (Figure 6B and Appendix A). It has been shown that thiamethoxam has an inhibitory effect on *N. ceranae* proliferation in the stingless bee *Melipona colimana* [54]. However, in *A. mellifera*, the effects of neonicotinoids on the parasite’s success seemed variable [59,60,61]. In all of these cases, the combination of neonicotinoids and *Nosema* led to increased mortality. Moreover, other deleterious effects of thiamethoxam on the physiology and behaviors of honey bees have been described [24,62,63,64]. Thus, even if thiamethoxam may partially restore the microbiota of infected bees, it cannot be seen as beneficial for the holobiont.

The present work shows that *P. acidilactici* can offset the strongly altered gut microbiota of *N. ceranae*-infected bees. However, it is worth noting that such a restoration of the microbiota does not necessarily infer improved honey bee health. To prove the probiotic potential of *P. acidilactici*, it is necessary to assess its effects on *A. mellifera* colonies.

## Figures and Tables

**Figure 1 microorganisms-12-00192-f001:**
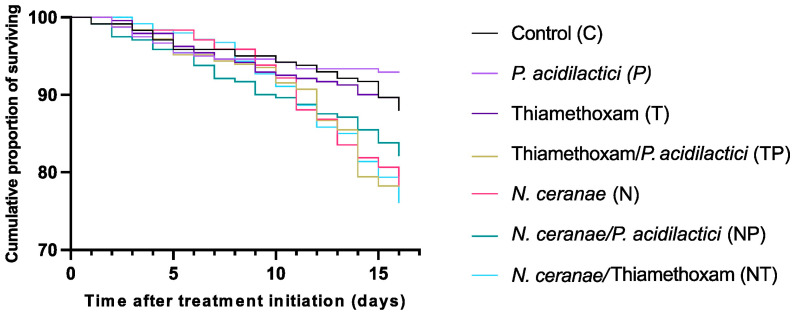
Cumulative proportion of survival of honey bees infected by *Nosema ceranae* (N), exposed to thiamethoxam (T) at 1.5 ng/mL, and exposed to *Pediococcus acidilactici* (P), alone or in combination. The Kaplan–Meier estimator was calculated on pooled replicates (log-rank Mantel–Cox test).

**Figure 2 microorganisms-12-00192-f002:**
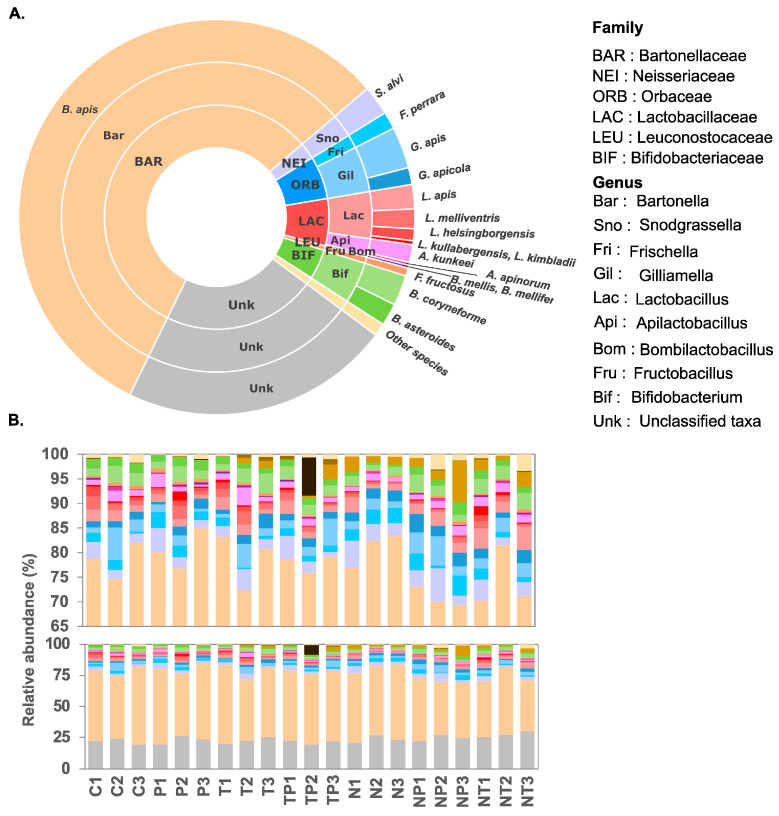
(**A**) Average taxonomic composition of the gut microbiota of control bees from family to species (n = 3). Core microbiota, non-core microbiota, and unknown represent 14.1 ± 2.5%, 64.0 ± 4.8%, and 21.9 ± 2.4%, respectively. (**B**) Taxonomic composition from each sample at the species level, control (C1, C2, C3), *Pediococcus acidilactici* (P1, P2, P3), thiamethoxam (T1, T2, T3), thiamethoxam and *P. acidilactici* (TP1, TP2, TP3), *Nosema ceranae* (N1, N2, N3), *N. ceranae* and *P. acidilactici* (NP1, NP2, NP3), and *N. ceranae* and Thiamethoxam (NT1, NT2, NT3). Same color as in A except for *Proteus* spp. (*P. mirabilis* and *P. penneri*) (light brown), *Providencia* spp. (dark brown), and *Arsenophonus* spp. (black).

**Figure 3 microorganisms-12-00192-f003:**
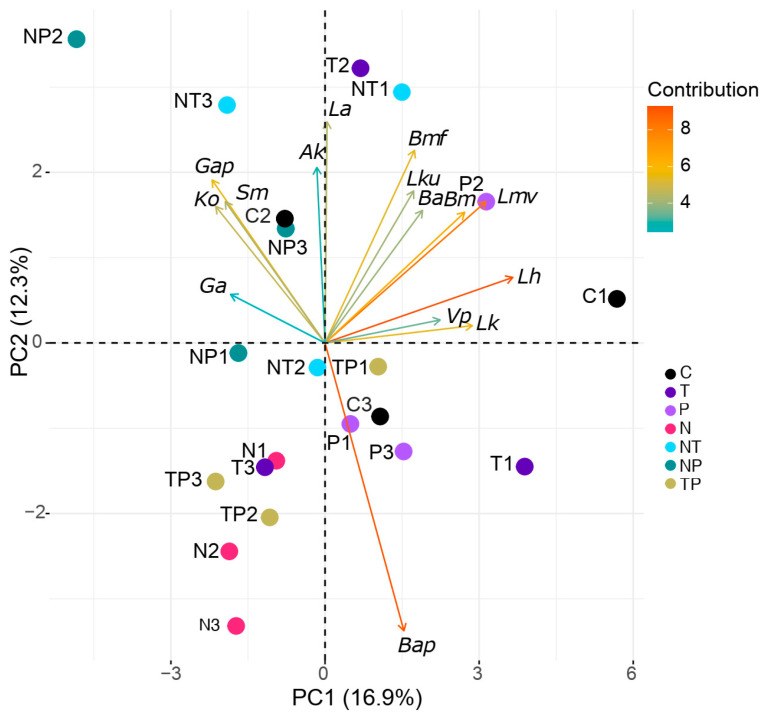
Unsupervised global PCA from bacterial species relative abundance combining all conditions (n = 3). Control (C, black), thiamethoxam (T, purple), *Pediococcus acidilactici* (P, mallow), thiamethoxam and *P. acidilactici* (TP, gold), *Nosema ceranae* (N, pink), *N. ceranae* and *P. acidilactici* (NP, green), and *N. ceranae* and thiamethoxam (NT, cyan). The arrows indicate the 15 most explicative variables: *Ga*: *Gilliamella apicola*, *Ko*: *Klebsiella oxytoca*, *Gap*: *Gilliamella apis*, *Sm*: *Serratia marcescens*, *Ak*: *Apilactobacillus kunkeei*, *La*: *Lactobacillus apis*, *Bmf*: *Bombilactobacillus mellifer*, *Lku*: *Lactobacillus kullabergensis*, *Ba*: *Bifidobacterium asteroides*, *Bm*: *Bombilactobacillus mellis*, *Lmv*: *Lactobacillus melliventris*, *Lh*: *Lactobacillus helsingborgensis*, *Vp*: *Virgibacillus proomii*, *Lk*: *Lactobacillus kimbladii*, and *Bap*: *Bartonella apis*.

**Figure 4 microorganisms-12-00192-f004:**
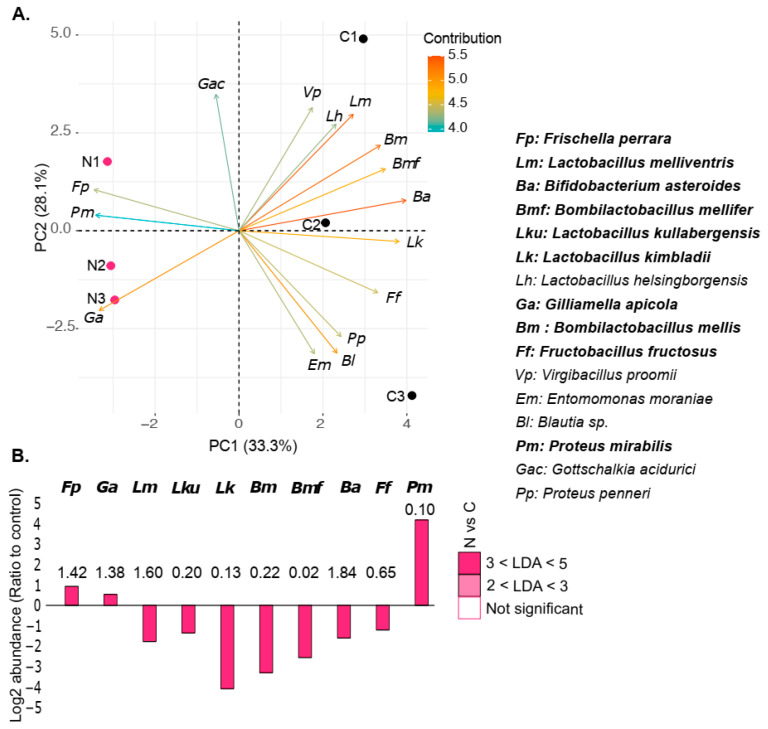
Principal component analysis between C and N (PCA). (**A**) Based on the relative abundance (Metaphlan4) of the species in the gut microbiota infected by *N. ceranae* (N, pink) or the not-infected control group (C, black). (**B**) Mean log2 ratio of abundance in infected relative to control honey bees of species significantly enriched (positive values) or depleted (negative values) in response to the parasite (n = 3). The values above the bars represent the average species abundance in the control honey bees relative to the total relative abundance in percentage. Species in bold are significantly different compared to the respective control (N vs. C).

**Figure 5 microorganisms-12-00192-f005:**
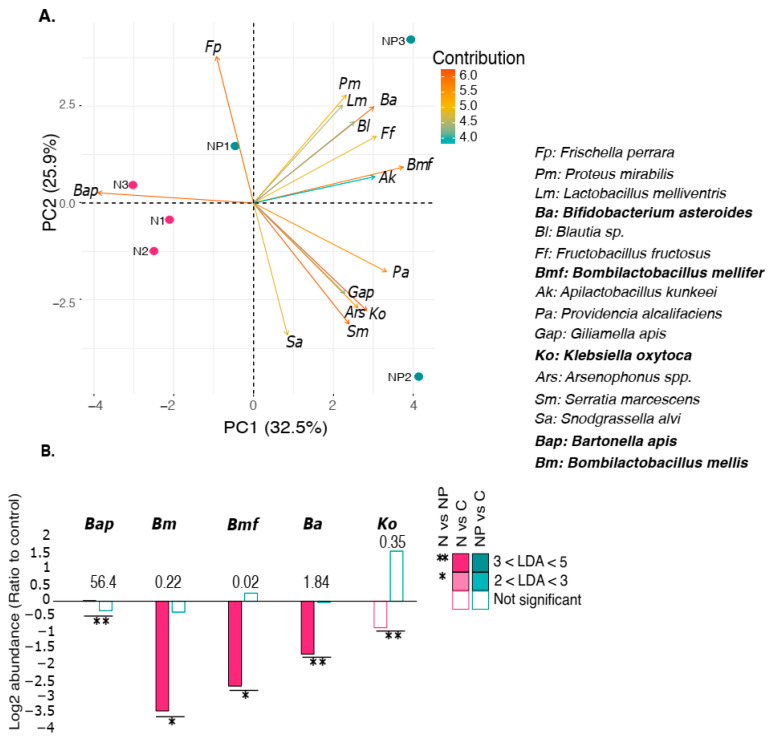
Principal component analysis (PCA) between N and NP. (**A**) Shows the clustering based on the relative abundance (Metaphlan4) of the species in the gut microbiota between the group infection by *Nosema ceranae* (N, pink) versus the group infection by *N. ceranae* and fed with *Pediococcus acidilactici* (NP, green). (**B**) Mean log2 ratio of abundance in *N. ceranae* (N) and *N. ceranae* and *P. acidilactici* (NP) relative to control honey bees of species significantly enriched (positive values) or depleted (negative values) under either condition (n = 3). The values above the bars represent the average species abundance in the control honey bees relative to the relative abundance in percentage. *G. acidurici* is not on the graph bar because it is totally absent from the NP conditions, but it is significantly enriched in the N condition (LDA > 3). Species in bold are significantly different compared to the respective control (N vs. NP).

**Figure 6 microorganisms-12-00192-f006:**
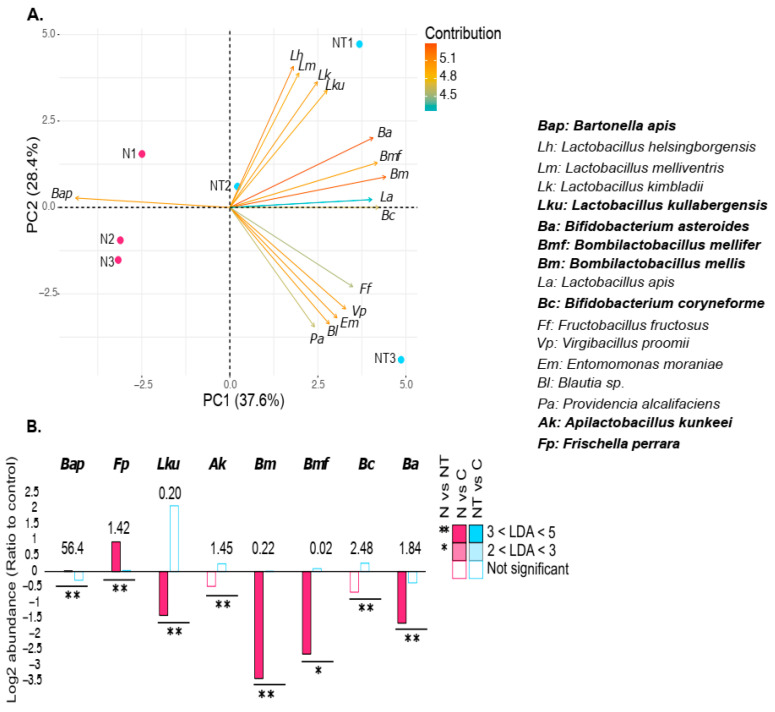
Principal component analysis (PCA) between N and NT. (**A**) Shows the clustering based on the relative abundance (Metaphlan4) of the 15 species in the gut microbiota that contribute to separating the samples between the group infected by *Nosema ceranae* (N, pink) and the group infected and exposed to thiamethoxam (NT, cyan). (**B**) Mean log2 ratio of abundance in *N. ceranae* (N) and *N. ceranae* and thiamethoxam (NT) relative to control honey bees of species significantly enriched (positive values) or depleted (negative values) under either condition (n = 3). The values above the bars represent the average species abundance in the control honey bees relative to the total microbial content in percentage. Species in bold are significantly different compared to the respective control (N vs. NP).

## Data Availability

Data are contained within the article and Appendix A.

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
