# Peer review of "The Response of the Honey Bee Gut Microbiota to Nosema ceranae Is Modulated by the Probiotic Pediococcus acidilactici and the Neonicotinoid Thiamethoxam"

_microorganisms, 2024, doi:10.3390/microorganisms12010192_

Round 1

Reviewer 1 Report

Comments and Suggestions for Authors

The present study tests how the microbiota of honey bees is modulated by N. ceranae, thiametoxam and the probiotic Pediococcus acidilactici, alone or in combination, and determines that the later modulates microbiota of bees infected with the parasite. Although the work is well presented and tests a very interesting question I find however a main issue concerning to the basal infection levels with Nosema in control bees which may blurry the results.

Authors refer to “low levels of basal infection” in control bees, however I don’t believe 35% of infection could be consider as low. Besides these basal levels of infection are not equivalent along the honey bee groups that didn’t receive Nosema spores. For instance, see the percentage of infection between groups C vs. TP for all colonies, or C vs T in colony Col2 and Col3. I believe that at least similar levels of infection should have been found along these groups, unless either the bacteria or the neonicotinoid had a direct effect on N. ceranae infection, which doesn’t seem to be the case given that the three colonies show different trends of infection levels between those groups. Whether the results presented highlight some modulation of the microbiota by either P. acidilactici or thiametoxan in Nosema-fed bees, given the basal % of infection I have doubts about how to discern between the effects produced when more spores are given. Also, taking a look at supp figure 1A, Col2 shows nearly a 45% of infection, which is almost half of the bees, being the infection levels much higher than other groups not receiving Nosema spores.

Regarding to mortality, the work shows that mortality of TP group is higher than the other groups not receiving Nosema spores. However, there is a high percentage of infection in this group, especially in Col3, which may explain this result.

Also, I am concern about the method they use to control for the presence/absence of Nosema ceranae. Given the similarity between the spores of Nosema ceranae and Nosema apis, how are they 100% sure their bees are also free of this second parasite. A PCR must be done instead.

Author Response

 First, we would like to thank you for your comments and suggestions that improved the quality of our manuscript. We have addressed each of the issues you raised in a point-by-point reply. We hope that new changes did in our manuscript will answer to all queries.

Comments 1. The present study tests how the microbiota of honey bees is modulated by N. ceranae, thiametoxam and the probiotic Pediococcus acidilactici, alone or in combination, and determines that the later modulates microbiota of bees infected with the parasite. Although the work is well presented and tests a very interesting question, I find however a main issue concerning to the basal infection levels with Nosema in control bees which may blurry the results.

Response 1. Thank you for your thorough reading. We understand the question raised by the basal infection prevalence regarding the survival analysis. This is why we have simply highlighted the inter-colony and inter-individual variability.

For microbiota analysis, we prevented this bias by verifying the presence or absence of Nosema spores in each bee sampled for dissection, as mentioned in the Material and Methods section. We were thus sure of the infection status of the individuals before sequencing.

Comments 2. Authors refer to “low levels of basal infection” in control bees, however I don’t believe 35% of infection could be consider as low.

Response 2. Following your comment, we eliminated the term "low" regarding basal infection (line 142).

Comments 3. Besides these basal levels of infection are not equivalent along the honey bee groups that didn’t receive Nosema spores. For instance, see the percentage of infection between groups C vs. TP for all colonies, or C vs T in colony Col2 and Col3. I believe that at least similar levels of infection should have been found along these groups, unless either the bacteria or the neonicotinoid had a direct effect on N. ceranae infection, which doesn’t seem to be the case given that the three colonies show different trends of infection levels between those groups. Whether the results presented highlight some modulation of the microbiota by either P. acidilactici or thiametoxan in Nosema-fed bees, given the basal % of infection I have doubts about how to discern between the effects produced when more spores are given. Also, taking a look at supp figure 1A, Col2 shows nearly a 45% of infection, which is almost half of the bees, being the infection levels much higher than other groups not receiving Nosema spores.

Response 3. The addition of thiamethoxam or P. acidilactici had no impact on the prevalence, nor on the quantity of N. ceranae (as seen by mapping the reads on the genome of N. ceranae). Yet, we agree that the prevalence of basal infection varied between cages or conditions, that could reflect a technical or random variation between cages. The modulation of the relative abundance of bacterial species was not linked to the quantity of N. ceranae nor to its prevalence. Moreover, as mentioned above, we sequenced bees that were verified to be non-infected bees (devoid of Nosema spores) for all conditions without Nosema (C, T, P, TP) and verified to be infected for other groups. Thus, the significant differences between non-infected and infected groups could be related to N. ceranae.

Comments 4. Regarding to mortality, the work shows that mortality of TP group is higher than the other groups not receiving Nosema spores. However, there is a high percentage of infection in this group, especially in Col3, which may explain this result.

Response 4. Col3 was the colony with the highest level of natural infection but the mortality (in the TP group) was higher in the Col2 colony which however presented a lower level of infection (see Fig. S1 and Table S1). So, this is not the (natural) infection that can explain the higher mortality observed in the TP group.

Comments 5. Also, I am concern about the method they use to control for the presence/absence of Nosema ceranae. Given the similarity between the spores of Nosema ceranae and Nosema apis, how are they 100% sure their bees are also free of this second parasite. A PCR must be done instead.

Response 5. We agree that the microscopic observation of spores was not sufficient to differentiate spores of N. apis and N. ceranae. Before performing samplings, the absence of N. apis in the colonies was checked by PCR. In parallel, the spores produced for infection were checked by PCR, showing the presence of N. ceranae and the absence of N. apis. Moreover, “we also verified the absence of N. apis by mapping all the reads on a 382-bp fragment of the N. apis gene encoding the small subunit ribosomal RNA. No read mapped this region.” We added this sentence in the Materials and Methods section (Lines 119-121).

Reviewer 2 Report

Comments and Suggestions for Authors

1The abstract should specify the results of the research, not just simply describe the results of the experiment.

2In the introduction, some endogenous and exogenous bacteria can act as probiotics to protect the health of bees, but the role of these bacteria has not been specifically demonstrated, and the function of the core intestinal flora of bees has not been described.

3In Method part:

Too many groups, too many variables, and not sure what your topic is? Pesticides? A parasite? Or probiotics? The experimental design that emerged was very messy.

4In line 113 What ca means? What is ad libitum means in line 118?

5I don't see the concentration of pesticide. How does the parasite infect its host? Is there a standard to define it? How is feeding done? The exact method is not described.

6For the Sampling method, Is the gut the whole gut or the hindgut or midgut?

7In the Results section:The picture is fuzzy, which can improve the resolution of the picture. What does the picture in Figure 1B represent? The author does not seem to explain it in detail in the figure, and the brown color in Figure 1B is not shown in Figure 1A. Which bacteria does the brown color represent in Figure 1B? The PCA results in Figure 2 show two groups of separation, which is suggested to be displayed on the graph using elliptical confidence intervals. If there is no difference in alpha and beta diversity, it can be placed in the supplementary material.

8Please explain the meaning of "at a high taxonomic resolution" in line 93

9What bees are collected in lines 98-110 (foragers, nurseries), are they healthy, and do they have other pests and diseases that are not stated in the text.

10Is Microsporidium mixed with Lactococcus and thiamethoxam, and what is the final concentration of both mixed feeding? The description in lines 118 to 121 is not clear. You are advised to modify it.

11Line123, the Test after 16 days, what is the basis for determining the number of days, why not other time or set a time periods of testing?

12Among the seven groups in Line 204-213, the most abundant genus was Bartonella, accounting for more than 50%. Bartonella was a non-core flora, indicating that bees were in sub-health conditions at this time, including the control group, which could not truly reflect the response of intestinal flora when bees were subjected to external stress. The conclusion can only be the changes of intestinal flora of bees under sub-health conditions. Or the authors may be able to provide intestinal microbiota data from local healthy western bees

13Line 96 reveals that Bartonella is more than 50% abundant in 7 groups, what its role is, and what effect it has on bees' resistance to external stress. The authors do not clearly describe the reasons for the high abundance of Bartonella in the seven groups.

15The purpose of this paper is to reveal the response of the flora to external stress, how the gut flora respondsin addition to the abundance change, what is the change of its metabolites? . The author determined the results after 16 days, and did not determine the changes of intestinal flora in different time periods, so it would be better to determine the corresponding changes of intestinal flora

16What is the significance of the seventh group set by the author? No authors describe the results in the paper?

Author Response

 First, we would like to thank you for your comments and suggestions that improved the quality of our manuscript. We have addressed each of the issues you raised in a point-by-point reply. We hope that new changes did in our manuscript will answer to all queries.

Comments 1. The abstract should specify the results of the research, not just simply describe the results of the experiment.

Response 1. In order to put more emphasis on the overall results of the experiment, we have modified the last sentence of the abstract (lines 35-38) as follows: "This study showed that stressors and probiotics may have antagonistic impacts on the honey bee gut bacterial communities and that P. acidilactici may have a protective effect against the dysbiosis induced by an infection with N. ceranae."

Comments 2. In the introduction, some endogenous and exogenous bacteria can act as probiotics to protect the health of bees, but the role of these bacteria has not been specifically demonstrated, and the function of the core intestinal flora of bees has not been described.

Response 2. We fully agree with your comments. We modified the sentence in the Introduction section as follows (line 74): “The use of endogenous and exogenous bacterial strains as potential probiotics has been suggested as a strategy to enhance the honey bee health by mitigating the detrimental effects on the gut microbiota of various pathogens, including N. ceranae, although their mode of action remains to be determined. »

Comments 3. In Method part:

Too many groups, too many variables, and not sure what your topic is? Pesticides? A parasite? Or probiotics? The experimental design that emerged was very messy.

Response 3. We understand that there were three treatments applied, alone or in combination, leading to numerous experimental groups (i.e. 7 groups including the untreated control bees). We believe that this apparent complexity will be reduced by clarifying the objectives of the paper at the end of the introduction (see lines 79-83): “This study aimed to decipher the impact of N. ceranae and thiamethoxam, alone or in combination on the honeybee gut microbiota, and to test whether P. acidilactici was able to counteract some alterations. Honeybee workers were infected with N. ceranae spores, orally exposed to a low dose of thiamethoxam and treated with P. acidilactici, alone or in combination. The modulations of the gut microbiota were studied at the species level using whole genome shotgun metagenomics.”. This led to seven experimental groups, as described in Materials and methods section (lines 91-94) : (1) untreated control (C), (2) treatment with P. acidilactici (P), (3) chronic exposure to 1.5 ng/mL thiamethoxam (T), (4) infection by N. ceranae (N), (5) infection by N. ceranae and treatment with P. acidilactici (NP), (6) infection by N. ceranae and chronic exposure to 1.5 ng/mL thiamethoxam (NT), (7) chronic exposure to 1.5 ng/mL thiamethoxam and treatment with P. acidilactici (TP).

Comments 4. In line 113 What ca means? What is ad libitum means in line 118?

Response 4. “ca.” is the abbreviation for circa which means “approximately”, “around”, “about”. “Ad libitum” means “without restriction”, “at will”, “as often as necessary”. It is commonly used in animal science to indicate that animals have permanent access to food. These two Latin terms are commonly used in scientific articles and should be italicized.

Comments 5. I don't see the concentration of pesticide. How does the parasite infect its host? Is there a standard to define it? How is feeding done? The exact method is not described.

Response 5. The concentration of thiamethoxam administered was 1.5 ng.ml-1 in sucrose syrup. This concentration is indicated in the Materials and Methods section (see lines 92 and 94) and to be clearer we added the concentration in the legend of figure 1 (line 156).

We also added some explanations about the parasite infectious cycle in the Introduction section (lines 62-63): “This ubiquitous parasite is transmitted by its spores via the feco-oral route between bees, particularly during cleaning activities.” We also included the corresponding reference: Martín‐Hernández R, Bartolomé C, Chejanovsky N, Le Conte Y, Dalmon A, Dussaubat C, et al. Nosema ceranae in Apis mellifera : a 12 years postdetection perspective. Environmental Microbiology. avr 2018;20(4):1302‑29.

As detailed in lines 96-97, “Nosema ceranae-infected groups were collectively infected by adding ca. 105 spores per bee in sugar syrup.” This method is commonly used to infect honeybees with this parasite, as described by Paris et al (2020). We added this reference. To be more precise, we also replaced the sentence “The syrup was consumed within 12 h” by “When the feeders containing N. ceranae spores were empty (within 12 hours), they were removed and replaced with feeders containing only sucrose syrup” (see lines 97-98).

Comments 6. For the Sampling method, Is the gut the whole gut or the hindgut or midgut?

Response 6. As specified in line 108 “…six whole guts, excluding crops, were dissected…”. We chose to remove the crop because it is often empty or detached during the dissection and because it may contain many exogenous environmental bacteria while almost all gut bacteria lay in the midgut and the hindgut.

Comments 7. In the Results section : The picture is fuzzy, which can improve the resolution of the picture. What does the picture in Figure 1B represent? The author does not seem to explain it in detail in the figure, and the brown color in Figure 1B is not shown in Figure 1A. Which bacteria does the brown color represent in Figure 1B? The PCA results in Figure 2 show two groups of separation, which is suggested to be displayed on the graph using elliptical confidence intervals. If there is no difference in alpha and beta diversity, it can be placed in the supplementary material.

Response 7. Figure 2B represents the relative abundance of all the species detected in each sample to have a view of the colony variability in each condition. The (light and dark) brown and black colors are not shown in the figure 2A because these bacteria were missing in honeybees from control group and mainly caused by the treatment, that’s why each colour was described in the legend of the figure : « Same color as in A except for Proteus spp. (P. mirabilis and P. penneri) (light brown), Providencia spp. (dark brown) and Arsenophonus spp. (black) ». (See lines 178-180).

The quality of the figure was fine when it was submitted and we will make sure it is adequate for publication.

The PCA (Figure 3) did not represent alpha and beta diversity but the relative abundance of species in each sample. This representation allowed us to determine which condition(s) could be interesting for further study. To be consistent with the other figures, we have chosen not to use ellipses on the PCA.

Comments 8. Please explain the meaning of "at a high taxonomic resolution" in line 93

Response 8. To be more precise we have replaced "at a high taxonomic resolution" by “a taxonomic resolution at the species level” in the abstract (line 29) and by “were studied at the species level using whole genome shotgun metagenomics” in the Introduction section (line 82).

Comments 9. What bees are collected in lines 98-110 (foragers, nurseries), are they healthy, and do they have other pests and diseases that are not stated in the text.

Response 9. Collected bees were “interior” bees taken from frames of three colonies in the warmest period of the day, when most foragers (corresponding to older bees) are visiting resources. It was then a mixture of bees of various ages.

Honeybees were apparently healthy since there was no high mortality in control conditions (Figure 1). Yet, we fully agree that colonies may have been exposed to other biotic (parasites, pathogens) and abiotic (pesticides, pollutants) stressors in their natural environment, which is a main bias when working with such environmental samples. Thus, we decided to make biological replicates by considering independent colonies to reduce the impact of such potential external factors.

Comments 10. Is Microsporidium mixed with Lactococcus and thiamethoxam, and what is the final concentration of both mixed feeding? The description in lines 118 to 121 is not clear. You are advised to modify it.

Response 10. We added the required precisions: “For the TP condition, thiamethoxam and P. acidilactici were given in different feeders at 1.5 ng/ml and 104 CFU/ml, respectively” (see lines 103-104).

The infection with N. ceranae was done once at day 0, after which the different treatments began. To emphasis this, we modified the description of the experimental conditions by precising the honeybees were infected “THEN” exposed to thiamethoxam or P. acidilactici. (see line 93).

Comments 11. Line123, the Test after 16 days, what is the basis for determining the number of days, why not other time or set a time periods of testing?

Response 11. Several studies have shown that N. ceranae induces high mortality 15 days after infection. We therefore chose 16 days to maximize the chance of seeing an impact on the gut microbiota. This also ensures that N. ceranae has completed several development cycles. Finally, in the context of chronic exposure to thiamethoxam, this time frame seemed relevant with regard to the bee longevity.

Due to the number of conditions and replicates it was not possible to perform a time dynamic of the microbiota. This may be of interest when focusing on a limited amount of conditions.

Comments 12. Among the seven groups in Line 204-213, the most abundant genus was Bartonella, accounting for more than 50%. Bartonella was a non-core flora, indicating that bees were in sub-health conditions at this time, including the control group, which could not truly reflect the response of intestinal flora when bees were subjected to external stress. The conclusion can only be the changes of intestinal flora of bees under sub-health conditions. Or the authors may be able to provide intestinal microbiota data from local healthy western bees.

Response 12. The presence of Bartonella apis does not necessarily indicate that the bees were in poor health. As stated earlier the low mortality of control bees showed that they were quite healthy. A high relative abundance of B. apis has been reported in other studies, including in old bees, winter bees, and in bees under low-protein diet. As mentioned in the discussion (line 268 to 275), bees were sampled in September, a period of the year that could explain this shift. Furthermore, other unpublished data on local honeybees showed such high abundance of Bartonella. We believe that this abundance of Bartonella is rather explained by location, sampling period and resources consumed by the bees than by health status.

Comments 13. Line 96 reveals that Bartonella is more than 50% abundant in 7 groups, what its role is, and what effect it has on bees' resistance to external stress. The authors do not clearly describe the reasons for the high abundance of Bartonella in the seven groups.

Response 13. The abundance of Bartonella may not be related to stress (see previous comment). The role of Bartonella in the gut has not been described and we cannot explain why its abundance is high in our colonies. As mentioned above, such high abundance has already been observed. However, the relative abundance of B. apis never changed significantly between the 7 groups, suggesting it is not impacted by the applied stressors.

Comments 15. The purpose of this paper is to reveal the response of the flora to external stress, how the gut flora responds?in addition to the abundance change, what is the change of its metabolites ?. The author determined the results after 16 days, and did not determine the changes of intestinal flora in different time periods, so it would be better to determine the corresponding changes of intestinal flora

Response 15. We fully agree that it would have been ideal to determine at many time points the modulation of metabolites, but also of the proteins and transcripts in the gut. This would have implied a much complex experimental design as well as more resources. Nevertheless, we agree that the present work showed conditions of interest that should be further investigated in future analyses on a restricted number of experimental conditions, as we intend to do.

Comments 16. What is the significance of the seventh group set by the author? No authors describe the results in the paper.

Response 16. Despite high mortality, the TP condition did not show much modulation of the microbiota, with the abondance of only two species, F. fructosus and B. mellifera, reduced by the addition of the probiotic. We included these data in the revised Table S2. According to the PCA analysis we chose to detail only the most modulated conditions.

Round 2

Reviewer 2 Report

Comments and Suggestions for Authors

No.